# A Review of Recent Developments in Edible Films and Coatings-Focus on Whey-Based Materials

**DOI:** 10.3390/foods13162638

**Published:** 2024-08-22

**Authors:** Arona Figueroa Pires, Olga Díaz, Angel Cobos, Carlos Dias Pereira

**Affiliations:** 1Polytechnic Institute of Coimbra, College of Agriculture, Bencanta, 3045-601 Coimbra, Portugal; arona@esac.pt; 2Department of Analytical Chemistry, Nutrition and Food Science, Faculty of Sciences, Food Technology Area, Campus Terra, Universidade de Santiago de Compostela, 27002 Lugo, Spain; olga.diaz.rubio@usc.es (O.D.); angel.cobos@usc.es (A.C.); 3Research Centre for Natural Resources, Environment and Society (CERNAS), Bencanta, 3045-601 Coimbra, Portugal

**Keywords:** food packaging, films and coatings, mechanical and functional properties, edible films, whey protein films and coatings

## Abstract

Packaging for food products is particularly important to preserve product quality and shelf life. The most used materials for food packaging are plastic, glass, metal, and paper. Plastic films produced based on petroleum are widely used for packaging because they have good mechanical properties and help preserve the characteristics of food. However, environmental concerns are leading the trend towards biopolymers. Films and coatings based on biopolymers have been extensively studied in recent years, as they cause less impact on the environment, can be obtained from renewable sources or by-products, are relatively abundant, have a good coating and film-forming capacity, are biodegradable and have nutritional properties that can be beneficial to human health. Whey protein-based films have demonstrated good mechanical resistance and a good barrier to gases when at low relative humidity levels, in addition to demonstrating an excellent barrier to aromatic compounds and especially oils. The use of whey proteins for films or coatings has been extensively studied, as these proteins are edible, have high nutritional value, and are biodegradable. Thus, the main objective of this document was to review new methodologies to improve the physicochemical properties of whey protein films and coatings. Importance will also be given to the combinations of whey proteins with other polymers and the development of new techniques that allow the manipulation of structures at a molecular level. The controlled release and mass transfer of new biomaterials and the improvement of the design of films and packaging materials with the desired functional properties can increase the quality of the films and, consequently, broaden their applications.

## 1. Introduction

The accumulation and slow degradation of fossil-based plastics in the environment cause serious ecological problems [1]. The global plastic packaging market is predicted to grow from USD 1098.81 billion in 2023 to USD 1333.02 billion in 2028 [2].

According to the Flexible Packaging Association, flexible packaging is the most used in food and contributes to more than 60% of the total packaging market [3]. The packaging of food products has a great impact on the environment, as it consumes a lot of raw materials, and used packages are not easily disposable. Due to the durability of plastics, design flexibility, lightness, and low price, these materials became advantageous over other raw materials. After processing, food products are generally packaged to prevent microbial spoilage and to preserve their organoleptic properties and their quality during their transportation and along shelf life [1,4,5]. Plastic films produced based on petroleum, such as polyethylene terephthalate (PET), high-density polyethylene (HDPE), low-density polyethylene (LDPE), polypropylene (PP), polyvinyl chloride (PVC), and polystyrene (PS), are widely used for packaging since they are easy to produce [6], widely available, have good mechanical properties and help to preserve the good characteristics of food (low permeability to water vapor, oxygen, carbon dioxide and aromatic compounds) [7]. However, traditional packaging materials present problems at the recycling level, are not degraded after disposal, cause many problems to the environment and wildlife, and use fossil fuel reserves, which are finite. Plastics also pose a series of risks and health problems when they enter the food chain as microplastics and may also be associated with the migration of harmful additives [8,9]. These reasons are the drivers to continue developing sustainable, fully biodegradable packaging materials, including edible films and coatings, providing a long shelf life for bioactive substances in foods and promoting safe packaging integration [9]. In this way, packaging materials obtained from biopolymers (proteins, polysaccharides, and lipids) represent a good alternative to petroleum-based materials to mitigate environmental problems since they are obtained from renewable sources or industrial by-products that are biodegradable [5,10]. As a result, films based on biopolymers have been extensively studied due to the low impact they have on the environment [11]. The biopolymer materials are being obtained from renewable sources or by-products and are increasingly being used in the food-packaging industry due to their relative abundance, film-forming capacity, biodegradability, and nutritional values [4,5,12]. They can be obtained from different animal and vegetable sources, extracted from biomass, synthesized from bio-monomers, or can be produced by microorganisms [13,14]. Some examples are the use of starch, cellulose, and collagen to produce biopolymer derivatives from waste food [15]. Due to their economic value, the most common biopolymers are polysaccharides (cellulose, starch, chitosan, pectin), and proteins (soy, gelatin, zein, sunflower) [5,16]. Therefore, the development of edible films and coatings from biopolymers is one of the promising strategies in recent years, as they allow replacing synthetic polymers that are very harmful for the environment. Particularly in the case of protein biopolymers, the use of whey protein-based films and coatings represents a promising solution since it allows for the valorization of this byproduct of the cheese industry. Besides, the films and coatings obtained from whey present excellent properties which will be highlighted in Section 3.

Figure 1 presents a diagram of the main biopolymers’ characteristics and how they are obtained.

## 2. Biopolymer Based Edible Films and Coatings

Edible films and coatings are an alternative to conventional plastics, with the main advantage being that they can be consumed together with food products without the need to be removed. They also have the potential to stabilize food products (extending their shelf life and maintaining the structural integrity of the food), are biodegradable, renewable, and mainly non-toxic, with minimal environmental impact [1,18,19].

Edible films and coatings have been used for many years to protect and extend the shelf life of many food products [20]. The use of coatings for food preservation began in the 12th century in China, where wax was used to protect fruit. The first edible films made from soy milk were used in Japan for fruit preservation and shine. However, due to the scarce variety of materials used, there was not much interest in these films. Methods such as refrigeration, controlled and/or modified atmosphere, sterilization or radiation, and smoking received more attention than edible packaging. Over the years, food preservation methods have increased and improved, offering opportunities to prepare, store, and consume food at any time of the year without altering the quality of the products [21].

Films and coatings can protect against ultraviolet light, transport solutes between food and the atmosphere (salts, additives and pigments), control the exchange of water vapor and gases (oxygen, carbon dioxide, nitrogen and ethylene), form a good protective barrier against mechanical damage, increase the shelf life of the product, may contain bioactive components such as antioxidants and probiotic or bioprotective microorganisms that are beneficial to health and can also have antimicrobial and antifungal effects [21,22,23]. Edible biodegradable films and coatings also reduce the rate of oxygen absorption, the migration of lipids, gases, and aromas and prevent the development of microorganisms during food storage [1].

### 2.1. Base Materials Used to Produce Biopolymers

As referred, the biopolymers used in the formulation of edible films and coatings are by-products from renewable agricultural resources, agrifood-waste or industrial by-products [1]. More recently, protein-based materials seem to be the most utilized since they provide very good nutritional value [24]. These materials are anti-allergic and biodegradable, non-toxic for human health, generating water, CO_2,_ and biomass under the action of physical factors and decomposing microorganisms [25]. Furthermore, edible films and coatings improve the shelf life of food as well as their safety, as they can carry active compounds within the film matrix or coating, such as anti-oxidants and antimicrobial compounds [26]. Some of these active compounds come in the form of natural product powders, essential oils, dyes, and flavorings and are incorporated into the film or coating-forming solution. These substances are generally consumed at the same time as the packaging material and also provide advantages to the final product as they improve its functional characteristics [25]. Films based on hydrocolloids and lipids are the most used materials, but films based on polysaccharides, due to the presence of many hydroxyl groups and hydrogen bonds, favor the formation of the film and are thus easier to obtain. Different properties can be observed between films and coatings made from negatively charged gums (alginate, pectin, or carboxymethylcellulose) [26].

The proteins used for the film’s formation are usually of animal origin (gelatin, casein, whey protein (WP), whey protein concentrates (WPC), whey protein isolates (WPI), collagen, or ovalbumin). However, plant-based proteins are important for a vegetarian diet (corn, soy, wheat, cottonseed, peanuts, and rice). For the formation of films and coatings, the protein is initially denatured by heat or pH adjustment or physical treatments such as UV radiation, followed by a conglomeration of peptide chains through new intermolecular interactions. Protein-based films are particularly suitable for meat products, as they have an affinity for the hydrophilic surfaces present in meat [21].

Unlike hydrocolloids, lipids (waxes, paraffins, glycerides) do not form cohesive films and are therefore used, especially for coatings or mixed with polysaccharides to obtain an optimized barrier to water vapor [21,22].

Depending on the biopolymers used to form edible films and on environmental conditions, such as humidity, these films may contain water of hydration that acts as a plasticizer by fitting into the polymeric chains, spacing them, and decreasing the transition temperature glass, thus improving flexibility. So, the water of hydration affects the mechanical and barrier properties of the film. Therefore, edible films are classified as hydrophilic, hydrophobic, or films containing both hydrophilic and hydrophobic components [23].

Hydrophilic films and coatings are made of polysaccharides and proteins that strongly interact with water, while hydrophobic films are formed from natural lipids, fats, and waxes. Since lipids have low polarity and, therefore, have a weak interaction with water, these films and coatings have good barrier properties to water vapor, whereas the barriers to oxygen and carbon dioxide are weak. Hydrophilic and hydrophobic films combine the advantages and lessen the disadvantages of each component. Hydrophilic components are a good oxygen barrier, and hydrophobic components are a good water vapor barrier. These films can be formed by a single layer or by multilayers. First, the layer of the film based on polysaccharides or proteins is formed, and then the lipid layer is applied. The properties of these films and coatings have been studied, and they show good barrier properties; however, separation and fracture of the lipid layer have been observed as a disadvantage [23].

Table 1 describes the principal biopolymers used and their functionalities for edible films and coatings.

Hydrocolloids obtained from animal or vegetable sources are widely used in food systems as gelling agents, thickeners, and stabilizers [28], and are obtained by the hydrophilic dispersion of colloids and are composed of long-chain polymers. They tend to swell when in contact with water and can partially or completely disperse, forming stable gels that show changes in their physicochemical properties, namely viscosity and thickness. The hydroxyl functional groups of polysaccharides allow them to be widely used in the preparation of hydrocolloid films due to their water solubility. In the case of proteins, they are also used to form hydrocolloid films, in some cases due to the structural hydrophilicity and intrinsic polydispersity of protein colloids. Combinations of synthetic and/or natural composites and polymers are also used, giving rise to binary films and coatings such as protein-protein, carbohydrate-carbohydrate, carbohydrate-protein, and lipid-based binary films [29]. Some examples of polysaccharides are alginate, agar, xanthan, carrageenan, guar gum, pectin, and locust bean gum [30]. Most polysaccharides used for edible films and coatings are cellulose derivatives, dextrans, inulin, alginate, carrageenan, starch derivatives, pectin derivatives, chitosan, seaweed extracts and galactomannans [9].

#### 2.1.1. Polysaccharide Based Films and Coatings

Polysaccharide films and coatings have good mechanical and oxygen barrier properties, good odor, and resistance to oils and fats, but the disadvantage is that they are permeable to moisture due to their hydrophilic nature [9,31]. The most used materials are cellulose and its derivatives (methylcellulose and hydroxypropylmethylcellulose) as these prevent the absorption of oils, alginic acid, that can be applied to meat products and greatly slows down lipid oxidation, chitosan (obtained from the exoskeleton of crustaceans and fungal cell walls and also from the deacetylation of chitin and which shows antimicrobial properties), starch and its derivatives since they are economical, easy to handle, odorless, light in color and tasteless and pectin, which has been used to produce films and coatings containing probiotics [21,22]. Cellulose is a very abundant organic compound that allows thermal gelatinization and forms water-soluble films through carboxymethylcellulose [32]. Starch, derived from various sources, is a polysaccharide consisting mainly of straight chains of amylose and highly branched chains of amylopectin and it is considered ideal for films and coatings formation [33]. Brown algae (*Phaeophyceae*) contain a polysaccharide called alginate with antimicrobial properties. Alginate has α-L-guluronate (G) and R-D-mannuronate (M) linkages in the (1–4) chain. Another polysaccharide with antimicrobial properties is pullulan, which is composed of maltotriose and α (1,6) glycosidic units, produced from starch by *Aureobasidium pullulans*. This polysaccharide is water soluble, colorless, odorless and tasteless, and forms edible films permeable to oils, being heat sealable [32].

#### 2.1.2. Lipid-Based Films and Coatings

The source and type of lipids used to produce films and coatings are oils and fats that come from vegetable or animal sources (cocoa, coconut, palm, peanut, butter, lard, fatty acids, and mono/di/triglycerides), waxes (beeswax, carnauba, jojoba, candelilla, and paraffin), natural resins (frankincense, guarana, and chicle) and emulsifiers and surfactants (fatty acids, fatty alcohols and lecithin). They are characterized by a high resistance to water penetration [21,31].

Lipids provide many features, such as gloss, minimize moisture losses, and reduce production costs and the complexity of packaging [34].

Edible lipid films have characteristics such as high barriers to water vapor and oxygen due to their relatively low polarity. However, due to this characteristic, they also present disadvantages when used as a protective coating on foods, as they can cause the appearance of rancidity, as well as a superficial coating of fat that can remain in food [35].

Generally, lipid films are mixed with hydrocolloids (cellulose, starch, proteins, polysaccharides) to obtain good film characteristics.

#### 2.1.3. Protein-Based Films and Coatings

Proteins are units formed by a covalent peptide bond found in different vegetable or animal sources [32]. Excellent barrier and mechanical properties make protein films a material with enormous potential for food packaging [17].

Protein’s ability to form films is strongly dependent on their molecular characteristics (molecular weight, conformation, charge, flexibility, and thermal stability). The degree of elongation, the nature of the amino acid sequences, and the distribution and sum of interactions between protein chains are closely related to the mechanical properties of the films [4].

Protein-based films have a high affinity for polar substances but have a low affinity for non-polar substances (e.g., oxygen, aromas, and oils). The addition of some other materials can reduce their sensitivity to humidity. However, permeability and humidity depend on the concentration and type of plasticizer used to form the film [36]. When compared to polysaccharides, proteins have lower water vapor permeability [21]. The use of whey proteins to make films has received a lot of attention since they are edible and biodegradable, allow for the valorization of byproducts from cheese manufacturing, have excellent mechanical properties, and are flexible, colorless, and odorless [12]. Section 4 deals specifically with whey-based edible films and coatings.

#### 2.1.4. Composite and Nanocomposite Films and Coatings

Importance is also given to the combinations of proteins with other polymers and to the development of new techniques that allow the manipulation of structures at the molecular level. Thus, through the controlled release and mass transfer of new biomaterials and improving the design of films and packaging materials with the desired functional properties, one can improve the films and, consequently, their applicability.

Composite films and coatings are formed by a combination of biopolymers that are used to improve the properties of films by mixing hydrophobic and hydrophilic materials to achieve the desired properties [29]. Composite films based on chitosan with lipids were studied by Wong et al. [37]. These films showed better efficiency in controlling moisture transfer when the lipid used was uniformly incorporated into the matrix. This feature demonstrated the importance of the morphological arrangement of the lipid within the chitosan matrix.

Composite films have a multilayer configuration [38]. According to the number of biopolymers combined, they are categorized as binary or ternary [29]. Protein-polymer composite films are studied to overcome the low water resistance of protein-based films and combine them with moisture-resistant biopolymers while maintaining the overall biodegradability of the final product [4]. The obtention of multilayer composite films tends to present some problems, such as layer delamination, time and energy consumption, and high production costs [39].

Nanocomposite films are multiphase materials containing a filler with at least one dimension less than 100 nm [40]. The use of bio/polymeric nanocomposites in the food packaging sector has increased in interest. Nanoparticles (NPs), when used in high proportions, are particularly interesting due to their high specific surface area, providing better reinforcement effects in the matrix [41]. The addition of nanoparticles to films or coatings can improve the mechanical, thermal, optical, and barrier properties of the films and coatings. However, possible problems related to the toxicity of these materials need to be evaluated in films (the use of nanocomposite films containing Ag NPs may be harmful to human health) [40,41].

These films are formed by a preformed thin matrix of a solution or by the dispersion of long-chain polymers. It is necessary to remove the solvent from the solution or dispersion to form the film matrix, thus reducing the distance between the polymers and favoring their interaction. This interaction causes an intertwining of the polymeric chains that increase the viscosity through the formation of a polymeric network [23]. Coatings are applied directly to the food product. Films can be produced from materials with film-forming capacity, and the components used are classified into three categories: hydrocolloids, lipids, and composites [42]. Even if they are not consumed with food, they can be more quickly and easily degraded when compared to plastic materials [21].

### 2.2. Plasticizers Used in Edible Films and Coatings

A plasticizer is defined by the International Union of Pure and Applied Chemistry (IUPAC) as “a substance or material incorporated in a material (usually a plastic or elastomer) to increase its flexibility, workability, or distensibility” [17].

Plasticizers are required for the formation of edible films and coatings, especially when polysaccharides or proteins are used as materials. The addition of plasticizers and emulsifiers to film-forming solutions aims to improve their mechanical properties or to increase stability when lipids and hydrocolloids are combined, prevents film breakage, and facilitates handling and storage, as they provide flexibility and elasticity by reducing intermolecular forces and increases polymer chain mobility [43]. For good film-forming, the type of base material, the plasticizers, the bioactive compounds, and other ingredients used are important factors to be considered [31]. Plasticizers for films can be defined as small, low molecular weight, non-volatile compounds added to polymers or proteins to reduce brittleness, provide flexibility, and increase film strength. As a specific definition for coatings, plasticizers reduce peeling and cracking, improving the flexibility and strength of the coating. In general, plasticizers reduce forces along polymer chains, thereby increasing free volume and chain movements [36]. Plasticizers are defined in two different ways: (1) plasticizers that are capable of forming hydrogen bonds and thus interact with polymers, interrupting the polymer-polymer bond and maintaining the distances between polymer chains and (2) plasticizers that interact with water, retaining more water molecules, and thus leading to higher moisture content and a larger hydrodynamic radius [44].

The most common plasticizers are polyols (glycerol, sorbitol, and polyethylene glycol), mono-, di-, or oligosaccharides (glucose, fructose-glucose syrups, sucrose, honey), as well as lipids and their derivatives (phospholipids, fatty acids, surfactants). Plasticizers have low molecular weight and are incorporated in film-forming materials as they lower the glass transition temperature [9,16,36]. An important parameter of hydrophilic plasticizers is the fact that they adsorb water, which affects the water barrier properties, leading to the destabilization of proteins. Thus, water plasticizes the hydrophilic phases, affecting the physicochemical properties and leading to possible changes in the properties of the material [45].

Protein-based films require the incorporation of plasticizers to reduce their brittleness, to allow easier removal from casings when formed, and to support and impart plastic properties. Plasticizing molecules lead to decreases in intermolecular forces along the polymeric chains, thus improving the flexibility, extensibility, tenacity, and shear strength of the film, but have the disadvantage of decreasing its mechanical strength and barrier properties [12]. Glycerol produces the best effects on whey protein films, leading to more stable, flexible, and less brittle films under various relative humidities [12]. Most plasticizers have hydroxyl groups that form hydrogen bonds with biopolymers and increase the free volume and flexibility of the film matrix. Plasticizers have different numbers of hydroxyl groups and are in different physical states (solid or liquid), which causes them to change the degree of rigidity and softening of the film [36].

## 3. Edible Protein-Based Films and Coatings’ Properties

The tensile strength (TS), Young’s modulus (YM), and elongation at break (EAB) are some of the mechanical properties that are necessary for edible films to have good application potential [46]. The measurement of water vapor permeability (WVP) in the films allows quantifying the amount of water that permeates per unit area and time (kg/m.s.Pa). WVP is also an important property of food packaging because it affects the shelf life of products [47,48].

Films based on WPI or WPC demonstrated better WVP than competing protein-based plant films (corn, zein, and isolated soy protein), and according to Ramos et al., they can be comparable to synthetic polymer films available on the market (Table 2) [49]. However, their mechanical characteristics suffer some limitations (TS, EAB, and YM values are lower when compared with plant protein-based films) [44,49]. WPI films are stronger and more flexible when compared with WPC films [48,50]. Protein films show better gas barrier properties, have specific mechanical characteristics, and have a greater intermolecular binding capacity when compared to films based on polysaccharides and lipids [43].

The properties of films and coatings are very important as they control the transmission of water vapor and gases, allowing for safety and prolonging the shelf life of food. To obtain films with good properties, they must have good flexibility, optical transparency, thermal stability, mechanical resistance, biodegradability, and high gas barrier properties [51]. For consumers, transparency and sensory properties are very important, but recently, functional properties have also become esteemed [52].

The low WVP of a polymeric material is important because it prevents dehydration and the absorption of moisture from the environment into the food product [30]. Relative humidity, temperature, and water resistance are also extremely important and are affected by film and coating thickness [53]. Mechanical properties of films, including tensile strength (TS), percentage of elongation at break (EAB), and Young’s modulus (YM), depend on the composition of the film and the nature of its components [51]. TS represents the maximum stress the film can support before breaking. EAB is the film’s ability to resist changes in shape without breaking. YM demonstrates the stiffness of the material and the flexibility of the film. These properties allow the film to resist external stresses, maintaining its integrity and thus allowing food to be protected [30]. However, mechanical properties are influenced by the type of material used and the manufacturing method of films and coatings [22].

## 4. Whey Protein-Based Films and Coatings

### 4.1. Methods to Obtain Whey Protein Films and Coatings

A film or coating is made by a combination of biopolymers and additives placed in an aqueous medium and must have a thickness of less than 0.3 mm [22]. The main difference between a coating and a film is in its application. Films and coatings solutions preparation is similar, but coatings are applied to the surface of the food products by different techniques like dipping, spraying, or brushing, whereas films are applied by wrapping the food (Figure 2). However, in both edible films and coatings, rigid matrices are formed with similar functions and similar characteristics.

The techniques used to develop films are like those used in the processing of synthetic plastics. In both cases, solvent casting (wet technique) and compression molding or extrusion (dry technique) are used. Biopolymers are generally compressed, molded, or extruded, generating a thermoplastic [54]. In these processes, native proteins are denatured through heating, the addition of acid or base, or a solvent that allows greater extension of their structure. Greater extension of the protein’s structure allows for greater interaction and greater association between protein chains (through electrostatic interactions, Van der Waals forces, hydrogen bonds, and covalent, and disulfide bonds) and thus generates a cohesive matrix of the protein film [4].

The formation of edible films is similar. Edible films can be formed by three mechanisms: (1) simple coacervation when a desolvation agent is added to produce the phase separation; (2) complex coacervation, which results from electrostatic interactions between oppositely charged polysaccharides and proteins, leading to the formation of a phase enriched in both polymers and another enriched in a solvent. The formation of this complex is influenced by different physicochemical parameters, such as pH, total concentration of biopolymers, ionic strength of the medium, proportion of biopolymer mixture, temperature and stirring time; (3) gelation or thermal coagulation is a simple method that relies on the complexation of positively charged polymers when in contact with specific polyanions to form inter- and intramolecular cross-links and form hydrogel granules, or negatively charged polymers in contact with polycations (e.g., calcium is the cation most used for gelation of most negatively charged polymers) [1,23].

Cheese whey (CW) and second cheese whey (SCW) are the by-products resulting from cheese and whey cheese production. CW and SCW have been proven to contain potential ingredients for the development of new food products like yogurts and ice cream with improved nutritional characteristics and other functionalities. Nowadays, due to their nutritional value, CW and SCW products have gained a noticeable position among healthy food products [55,56,57].

Whey protein films are generally obtained from whey protein isolates (WPI) and whey protein concentrates (WPC) [12]. WPI, WPC, and second cheese whey protein concentrate (SCWPC) are normally obtained with the aid of tangential filtration technologies. For WPI and WPC production, generally, the most popular techniques for whey preconcentration are ultrafiltration (UF) and/or diafiltration (DF), as well as other membrane processes such as nanofiltration (NF), reverse osmosis (RO), electrodialysis (ED) and microfiltration (MF). For their production, a UF/DF process is applied, and subsequently, the concentrated whey is pasteurized, evaporated, and dehydrated [56].

WPI has proteins with hydrophobic and thiol groups placed in the globular structure. Heat-induced denaturation of such proteins alters the 3D network, opening their structure and exposing sulfhydryl and hydrophobic groups that interact with other molecules and form strong covalent disulfide intermolecular bonds [58]. Commercial WPC and WPI contain about 35–80% and 90% protein, respectively, and their main components are β-lactoglobulin and α-lactalbumin, which are easily dissolved in water. WPI and WPC differ in their protein levels as well as in the levels of other constituents such as lipids, minerals, and, particularly, lactose. These differences strongly influence the intermolecular bonds of the films produced from WPC and WPI and, consequently, affect their barrier, mechanical, and thermal properties [12,59].

The edible whey film is formed by a highly interactive dry polymeric network and has a three-dimensional gel-like structure. Depending on the techniques for forming these materials, the final films can result in a spatially rearranged gel array that includes all additional film-forming agents. These films are obtained through the casting method, which consists of pouring the solution onto a flat surface to produce a dry gel and thus applying this film as an envelope to food products [10]. In the case of coatings, food products are immersed in the film-forming solution (approximately between 30 and 60 s) to ensure complete surface exposure with good adhesion and perfect integrity.

Both films and coatings based on whey are flexible, colorless, odorless, and transparent. Whey films possess some distinguishing features, such as their amphiphilic nature and electrostatic charges, as well as other factors, such as charge density and hydrophilic-hydrophobic balance, which can modify their conformation, as these factors define the physical and mechanic properties of films and coatings [10].

They have good physical properties at low relative humidities, are impermeable to oxygen, and have a good aroma but have high water vapor permeability due to their hydrophilic nature. This property can be significantly improved by combining these films with hydrophobic materials such as essential oils (EOs) or plant extracts, which have been shown to reduce WVP by increasing the content of hydrophobic groups [5,28]. Another advantage of whey-based films is that they are easily dissolved in water and can be prepared using water as a film-forming solvent [59]. The transparency, flexibility, and lack of odor and taste of these films favor their acceptance by consumers [10].

Whey proteins are β-lactoglobulin (β-Lg), α-lactalbumin (α-La), serum albumin (SA), immunoglobulins, lactoferrin, lactoperoxidase and enzymes (approximately 60), in addition to other nitrogen components, such as glycomacropeptide or caseinomacropeptide, which is released from κ-casein in the first step of enzymatic coagulation [56]. As referred, the properties of proteins make them excellent for forming films and coatings [60]. The amino acid sequence of proteins determines the interactions between the protein chains themselves, as well as with other components of the film [60]. For whey proteins, in particular, the thiol groups contained in the cysteine residues are capable of forming disulfide bonds [61]. β-Lg is the principal protein responsible for aggregation and gelling behavior [56] of whey proteins. α-La is the second largest fraction of whey proteins with a molecular weight of 14.2 kDa. The polypeptide chain is composed of 123 amino acids and eight cysteine residues and has a higher thermal stability as compared to β-Lg. Whey proteins’ ability to change chain conformations and interact with each other to form modified three-dimensional networks makes them excellent for use in films and coatings [61].

The process for the formation of the whey protein films is the same as used for other protein film formation. The first step in preparing a film is the unfolding of the protein’s native state, relieving low-energy intermolecular bonds. Protein unfolding and dissociation can be caused by various treatments, such as temperature change, pH change, shear forces, or the addition of organic solvents or salts [61]. For whey films and coatings, a solvent is necessary to dissolve WPI or WPC (5–12% in solution), generally in distilled water, and adjust the pH of the solution (normally to 7 or 8). This step is named the “solvent process”, where proteins are dispersed and solubilized in a solvent, such as water or ethanol [10,62]. Plasticizers are then added to the film or coating forming solution Then it is necessary to heat the solution (80–90 °C, 10–30 min) for protein denaturation. The process of thermal denaturation (“thermoplastic process”) involves several phenomena. First, at a temperature greater than 40 °C, β-Lg dimers dissociate into monomers. Denaturation occurs at 65 °C and unfolds the β-Lg molecule, exposing hydrophobic and thiol groups, which bond to other smaller β-Lg-containing aggregates or to other thiol-containing proteins. These smaller aggregates then interact to produce high molecular weight irreversible aggregates. The temperature for the irreversible denaturation of β-Lg is 69 °C and for α-La is 80 °C. β-Lg acts in the polymerization of whey proteins and defines their degree of denaturation. The aggregation rate is also affected by the free sulfhydryl groups that exist on β-Lg. In these films, other ingredients can be added before or after heating. Ingredients that can be heated (prebiotics, starches, blends) are included early in the process, while heat-sensitive ingredients (antioxidants, antimicrobial compounds, probiotics) are included after heating. Whey films that are produced without heat treatment break very easily during the drying process due to weak intermolecular interactions that occur [10]. Figure 3 presents examples of the production of films and coatings with milk proteins.

### 4.2. Advantages and Disadvantages of Whey Films and Coatings

The barrier properties of whey protein films are superior to those of polysaccharide films. However, these protein films have limitations regarding their mechanical characteristics since the molecular weight of the whey proteins has a strong influence on the fragility of the film. Thus, the use of some additional structuring agents, such as plasticizers to avoid fragility, emulsifiers to stabilize the base emulsion, lipids to increase WVP, and polysaccharides to improve barrier properties, is widely adopted [10]. Whey protein edible films and coatings use plasticizer content values between 10% to 60% (*w*/*w*). These levels depend on the properties of the film and the type of plasticizer used. Generally, the proportions of plasticizers in concentrated whey protein films are lower since non-protein components such as fat and lactose can interfere with film formulations [10]. Glycerol used as a plasticizer at very high levels in whey protein films improves moisture content solubility and water vapor barrier but reduces mechanical strength, Young’s modulus, and glass transition temperature [10]. The evaluation of the effect of two different plasticizers (glycerol and sorbitol) on the mechanical and barrier properties of whey protein films was conducted by Anker et al. [63]. The moisture content of WPI/glycerol films decreased until the 45th day and then remained constant until the 120th day. This affected the mechanical properties (increased stress at break, decreased strain at break, and increased glass transition temperature). Barrier properties were not affected.

Shaw and coworkers [64] evaluated the effects of glycerol, xylitol, and sorbitol on the physical properties of whey protein isolate films. The authors found that increasing the glycerol or sorbitol content led to increases in moisture content, water vapor permeability, and elongation but led to decreases in tensile strength, elasticity, and glass transition temperatures of the films. Increasing the xylitol content had no effect on permeability, moisture content, or glass transition temperature but decreased the elongation, tensile strength, and elasticity of the films. They also concluded that the moisture content of the films correlated well with the glass transition temperatures and that the differences in the physical properties of the films, depending on the type and concentration of the plasticizer, can be attributed to differences in the hygroscopic and crystalline properties of the plasticizers.

A study conducted by Huntrakul et al. [45] evaluated the effects of plasticizers glycerol, xylitol, sorbitol, polyethylene glycol 400, and oleic acid on water sorption, phase transitions, and other physical, mechanical, barrier, and stability properties of WPC and of mixed WPC-carboxymethilcellulose films. The effects of water sorption at 50% and 75% relative humidity on film stability were also studied. The results demonstrated that the use of plasticizers to modify the properties and stability of the bioplastic film based on mixtures of proteins and polysaccharides has a positive impact on film properties [45]. Mali et al. [65] evaluated the effect of plasticizer type (i.e., glycerol, sorbitol, and a 1:1 glycerol: sorbitol mixture) and concentration (0, 20, and 40 g/100 g of starch) on moisture sorption of films of cassava starch and concluded that films that did not contain plasticizer had the lowest monolayer value, while films containing 40 g of glycerol had the highest monolayer value. This value indicates the maximum amount of water that can be absorbed in a single layer per gram of dry film, as it is a measure of the number of sorption sites in a film. The addition of plasticizers originates more active sites, exposing the hydroxyl groups where water molecules are absorbed. The plasticizers also modified the starch network, making it less dense and facilitating the movements of the polymeric chain when the films were subjected to tension tests [65].

Perez et al. [18] developed and characterized edible films produced from WPC and plasticized with different levels of glycerol and/or trehalose to evaluate new formulations for edible films. In addition, changes in the mechanical properties of the film during storage at room temperature and under freezing conditions were tested. The authors concluded that both films have good transparency, as the addition of trehalose in the film formulation proved to be effective in preventing the Maillard reaction that darkens WPC-based films and can potentially generate toxic by-products, which may cause the loss of nutritional value of whey constituents of the edible film. However, they also found that the use of trehalose did not significantly improve the physicochemical properties of the films during long-term storage at different temperatures.

### 4.3. Improvement of Protein Film’s Properties

Various methods are being studied to improve protein films and make them stronger and less permeable. Modification of proteins (denaturation), use of other macromolecules (polysaccharides and lipids), use of bioactive compounds (microbial agents, probiotics, antioxidants), and plasticizers are some examples [66].

Cross-linking by the formation of stronger intermolecular covalent bonds in protein networks closer to the molecule’s packaging and the reduction of polymer mobility are methods that are used in the formation of protein films. Protein cross-linking can be done through heating, acid/base treatments, the addition of aldehydes, phenolic compounds, CaCl_2_, enzymes, irradiation, and ultrasonic/microwave-assisted treatment [4].

The functional properties of films and coatings can be improved through chemical, physical, or enzymatic crosslinking. A long chain structure with low permeability and greater tensile strength should result through increased protein interaction with chemical treatments. Enzymatic cross-linking techniques using enzymes such as lipoxygenase, polyphenol oxidase, transglutaminase, and peroxidase have been used to cross-link proteins. In this case, a specific enzyme, known as transglutaminase, capable of catalyzing covalent crosslinking processes between proteins, forms high molecular weight biopolymers [67]. The use of γ-radiation or UV radiation modifies the conformation of proteins, oxidizes amino acids, breaks covalent bonds, and produces free radicals within the protein, being able to originate proteins with greater molecular weight through electrostatic and hydrophobic interactions, or by the creation of disulfide bonds and interprotein cross-linking reactions [59,60,61]. Combining lipids and proteins to create a continuous, cohesive network improves the performance of films and coatings. The lipids disperse within the hydrocolloid matrix, forming emulsified films, or they can create a layer over it, leading to the development of bilayer films. Emulsified films received great attention through the “microvoid model”, according to which the transfer of gases and vapors occurs through microvoids that emerge during the drying of the emulsion, forming a hydrocolloid matrix. Another way is the “microvia model”, in which mass transfer occurs through the highly polymeric matrix itself. Thus, the addition of lipids to films and protein coatings gives flexibility to the film and can interrupt interactions between polymer chains [67].

However, these methods do not change the sensitivity to humidity that protein films have since it is not possible to replace all reactive sites. Another alternative method for improving the barrier and mechanical properties of protein films is being developed by combining the properties of several different polymers (composite films) [4]. Protein-Protein Composite Films [59], Protein-Polysaccharide Composite Films [68], Protein-Lipid Composite Films [62], Protein-Polymer Composite Films [69], Protein-Based Composite films [70] and Nanocomposite Films [40] are the composite films that are being studied.

Blends of proteins for the development of biodegradable films allow for an improvement in their physical and/or mechanical properties when compared to isolated protein films [4]. Jiang et al. investigated the microstructural and physical properties of WPI and gelatin composite films. The authors observed the formation of a compact film in which shrinkage of gelatin molecules was observed at the WPI/gelatin ratio 1:1. However, this aggregation process made the microstructure of the film discontinuous and led to lower perforation resistance of the film. The deformation and WVP of the composite films increased with the gelatin content, but significant variations in the moisture content and solubility were not observed [71].

The improvement in the mechanical properties of protein-polysaccharide films is attributed to the strong intermolecular interactions (hydrogen bonding, dipole-dipole bond formation, and charge effects) between hydroxyl groups of the polymer chains. Furthermore, cross-linking and heat treatment allow bonds between protein and polysaccharide chains (via Maillard reactions) and thus improve the mechanical properties of the polymer network [4]. Chakravartula and coworkers [72] studied edible films composed of different proportions of pectin, alginate, and WPC. The matrices showed good film formation capacity, except for whey protein at a certain concentration, with thickness, elastic, and optical properties correlated to the initial viscosity of the solution. Whey protein reduced the viscosity of the starting solutions compared to alginate or pectin solutions and decreased mechanical strength as well as water affinity. Pectin influenced the yellowing index, while alginate and whey protein affected the opacity of the film. Whey protein favored opacity, gas barrier properties and originated dense structures, resulting from polysaccharide-protein aggregates. All films showed good thermal stability.

The moisture barrier properties of protein-based films can be improved by adding lipids to their formulation and improving their mechanical properties. The addition of hydrophobic compounds to protein-based film formulations such as waxes, fats, oils, and fatty acids is important in such films. Protein-lipid composite films consist of a continuous phase (protein matrix) and a lipidic dispersed phase typically produced by emulsification or thin layer-by-layer lamination. In the case of emulsification, lack of homogeneity within the film matrix due to differences in lipid droplet size distribution is a problem. Thin layer-by-layer lamination faces problems in terms of cracking and/or delamination [4].

In the case of films and coatings produced with denatured whey protein, hydrogen, and disulfide bonds play a major role in their formation. While using whey protein in its native form, only hydrogen bonds contribute to the formation of films and coatings. Therefore, the covalent cross-linking produced by thermal denaturation in whey proteins is responsible for the insolubility, increased mechanical resistance, and O_2_ barrier properties of the film without changing the WVP of the film [10]. Denatured whey protein films are insoluble in water since their solubility decreases with temperature. They have higher tensile values, but the WVP remains identical when compared to native whey protein films. The latter are soluble in water and have lower traction values. Its opacity is also higher than the opacity of denatured whey protein films. Thus, increasing time or temperature during the thermal treatment of the film-forming solutions greatly improves firmness, robustness, and extensibility [10].

Another way to improve protein-based films is the use of nanocomposites. Polymeric nanocomposites exhibit improved packaging properties due to their nanosized dispersion [35,41]. Due to the nanometric size of nanocomposite particles, application in biopolymers leads to improvements in mechanical and barrier properties. The mechanical properties of nanocomposites induce large increases in the material’s stiffness (Young’s modulus) and can delay the passage of gases and water vapor, improving barrier properties. Another advantage is that nanocomposites do not affect the transparency of the films since their dimensions are smaller than micro and macro composites [73]. An important advantage is that biologically active ingredients can be added to impart the desired functional properties to packaging materials resulting from nanocomposites, thus offering enormous potential for application in the active food packaging industry [35].

## 5. Additives for Whey Based Functional Films and Coatings

Edible films with functional agents such as antimicrobials, antioxidants, and enzymes have also been objects of study and were used to improve the shelf life and quality of food products. The addition of different additives (organic acids, essential oils, plant extracts, antibacterial compounds, prebiotics, and probiotic microorganisms) in the films and coatings has a great interest in food preservation since oxidation and microbial deterioration are the main problems that affect food quality and safety [1,21,74]. Films and coatings can be used not only as protectors but also as carriers of bioactive substances (i.e., antimicrobial compounds, probiotics, anti-browning compounds, omega-3 fatty acids, and other nutraceuticals). Active food packaging with bioactive molecules acts as a traditional protection system and promotes consumers’ health [21]. Some related studies are summarized in Table 3.

### 5.1. Prebiotics and/or Probiotics Incorporated in Edible Films and Coatings

In recent years, consumers have been demanding healthier and natural foods, minimal processing, and the reduction or absence of additives. Foods containing probiotics confer health benefits when consumed adequately [75]. The recommended dose for human consumption of probiotic viable cells is 10^8–9^ per day [9,21].

Foods that naturally contain probiotics (e.g., *Lactobacillus acidophillus*, *Bifidobacterium*, and other probiotic *Lactobacillii*) or prebiotics can be used as dietary supplements, as they are good for combating lactose intolerance [9], increase resistance to invasion by pathogenic bacteria in the intestine [42], stimulate the immune system and also help against cancer [9,76]. The use of probiotics in foods that naturally contain them or by their incorporation has been increasing since they allow for the production of healthier and safer products. However, despite the benefits of these products, ensuring the viability of microorganisms during the processing and storage of these products and through their transit in the digestive tract has been a challenge for the industry. The number of probiotic microorganisms may decrease depending on their production and storage (oxygen, heat treatments applied, mechanical processes, and osmotic stress mechanisms). Prebiotics promote the growth of some microorganisms, aid in the absorption of minerals, and increase resistance against pathogens. The main sources of prebiotics are fruits and vegetables, the most common of which are fructooligosaccharides, galactooligosaccharides, and trans-galactooligosaccharides. Using prebiotics together with probiotics improves their viability [77].

The use of probiotics and prebiotics for food packaging is a very important innovation for the industry. Edible films and coatings incorporating probiotics and/or prebiotics represent a novel strategy that deserves further research work. Recently, edible films or coatings with probiotics have been the object of studies [42,75]. However, the application of probiotics in edible films is not an easy practice and represents a difficult task during food processing and storage. The techniques used for these processes must overcome osmotic and thermal stress and/or acid-induced stress that can occur during food preparation and storage [23].

The first study of films and coatings incorporating probiotics was carried out by Tapia et al. [78]. This study describes the use of alginate-gellan coatings and films for coating fresh apples and papayas through the incorporation of *Bifidobacterium lactis Bb-12^®^*. Viable probiotic cell numbers greater than 10^6^ CFU/g were maintained for 10 days of storage. Since then, many studies have been done on the incorporation of probiotics in edible films and coatings using different approaches. Pereira et al. [42] evaluated the stability of two probiotic microorganisms in edible films based on whey protein formulations. They verified that the WPI films did not suffer structural changes, thus being able to act as a good matrix in the incorporation of the bacteria *Bifidobacterium animalis Bb-12^®^* and *Lactobacillus casei-01* and verified that during 60 days of storage, they maintained adequate levels of viable cells. Other examples of the use of prebiotics and probiotics in whey-based films and coatings are presented in Table 3.

### 5.2. Antioxidants, Antibacterial and Antifungal Compounds

The addition of antioxidants to retard lipid oxidation in foods to improve the quality and extend the shelf life of foods (e.g., processed meat products) is necessary. Synthetic antioxidants such as butylhydroxyanisole (BHA), butylhydroxytoluene (BHT), and tert-butylhydroquinone (TBHQ) are widely used industrially. However, the use of these compounds has been questioned as raising safety issues. So, the use of spices and herbs as natural antioxidants could be an important advance for the food industry to protect food. Its effectiveness depends on several factors, such as dose and mode of application [79].

Akcan and coworkers evaluated the effectiveness of natural antioxidant extracts of laurel (*Laurus nobilis* L.) and salvia (*Salvia officinalis*) on edible films made of whey protein isolate applied to cooked meatballs over frozen storage at −18 °C for 60 days. This study showed that applying an antioxidant-active pack with laurel and/or salvia to meatballs effectively delayed oxidative changes [79].

Antimicrobial agents can influence the sensorial, mechanical, and optical characteristics and properties of edible films. Antimicrobial packaging materials are used as food preservatives because they improve the barrier properties of the packaging material and help increase the shelf-life of a food product by impeding microbial development. Furthermore, antimicrobial films and coatings have been proven to be capable of extending the lifecycle lag of microorganisms, reducing the rate of microbial growth, prolonging the shelf-life, and promoting the safety of food [35].

Essential oils (EOs) are natural antimicrobial compounds since, in nature, they protect plants from bacteria, fungi, and viruses, being used in food products to extend their shelf life. Generally, they are obtained from plants or spices, being excellent sources of biologically active compounds like terpenoids and phenolic acids [39]. Essential oils are extracted from distinct parts of the plant (roots, bark, leaves, seeds, and fruits) through steam distillation methods, mechanical processes, dry distillation for essential oils (EO), and solvent extraction for plant extracts. Essential oils and plant extracts have numerous safety advantages when compared to chemical preservatives and are approved by the Food and Drug Administration [80].

The antimicrobial activity of EOs is attributed to their phenolic content, mainly flavonoids and their derivatives. The mechanisms of antimicrobial activity of phenolic compounds are related to the destruction of the cell wall and cytoplasmic membranes, preventing the synthesis of DNA, RNA, proteins, and polysaccharides in bacteria and fungi [80]. Examples of essential oils such as cinnamon, bergamot, and lemon have been incorporated into edible films as these promote antimicrobial efficacy and improve the film’s barrier properties [81]. Many compounds have been studied as antimicrobial agents for food packaging, such as organic acids, enzymes, fungicides, and natural compounds (spices and essential oils) [82]. In previous works, our research group evaluated the potential of whey-based edible coatings to improve the shelf life of cheeses. In the first work [83], whey protein edible coatings were produced from ovine whey protein concentrate (WPC) with lactic acid and natamycin as antimicrobials. Two methods of coating polymerization were evaluated separately and in combination (heat denaturation and UV polymerization method). Microbiological analysis showed that coatings prevented the growth of *Staphylococcus* spp., *Pseudomonas* spp., *Enterobacteriaceae*, yeasts, and molds. With respect to external cheese evaluation, no differences were observed between cheeses concerning shape and rind color by visual inspection. Sensorial differences were found only for color homogeneity and hardness. It was observed that commercial-coated cheese had the lowest score in color homogeneity, whereas the cheese with the coating produced by HD polymerization was classified as the most uniform. Concerning the internal cheese evaluation, no differences were observed among cheeses regarding the color difference between paste and rind, odor, consistency, and flavor. Nevertheless, it was observed that for the first three attributes, the cheese bearing the commercial coating had the lower classification. In the work of Mileriene et al. [84], the bioactive edible coating formulated by integrating WPC and cinnamon extract adequately shielded the sensorial characteristics of curd cheese and inhibited the emergence of yeast and molds during its shelf life. The e valuated sensory properties (appearance, odor, taste, texture, and overall acceptability) of all samples were similar, indicating no effect of the coating on the flavor of curd cheese. More recently, second sheep’s cheese whey coatings containing essential oils (EOs) of oregano (*Origanum compactum*) and sage (*Salvia sclarea*) were tested in cheese. The physical, chemical and microbiological properties of these cheeses were studied over 28 days. The results indicated that cheeses coating with SCW:WPI (2:1) with or without EOs could successfully replace commercial coatings with the advantage of being edible packaging materials manufactured with by-products [85]. The use of EOs improved the film’s properties.

One example of a natural polycationic polysaccharide with antibacterial and antifungal properties is chitosan, which is present in marine crustaceans and produced by microorganisms (e.g., *Aspergillus niger*, *Mucor rouxii*, *Penicillium notatum*) or obtained from the alkaline deacetylation of chitin at high temperatures (above 80 °C). Chitosan has as its main properties biodegradability, non-toxicity, biocompatibility and antifungal properties, which make it very suitable for edible film formulations. Another very important function is the fact that it has a positive charge at biological pH, which allows complex coacervation by electrostatic interactions with a wide variety of proteins [1,73]. Many authors studied the potential application of edible films and coatings with chitosan [60,86,87,88].

Braber et al. [89] developed antifungal whey protein isolate (WPI) films with low amounts of a water-soluble derivative of chitosan. These films acted as a cross-linking agent through hydrogen bonds with whey proteins, thus reducing their solubility and elongation. The mechanical resistance and barrier to water vapor were maintained. These films showed excellent antifungal activity against *Aspergillus niger* and against the growth of other fungi that spoil food, namely *Fusarium* sp. and *Rhizopus* sp.

Another important antimicrobial agent with antioxidant and anticancer activities is carvacrol [23]. Carvacrol is found in essential oils like oregano (*Origanum vulgare*), thyme (*Thymus vulgaris*), peppermint (*Lepidium flavum*), wild bergamot (*Citrus aurantium bergamia*) and other plants. The antimicrobial activity of this compound is higher than most other volatile compounds present in essential oils, and this is due to the presence of the free hydroxyl group, its hydrophobicity, and its phenol fraction [90]. Whey protein nanofibers (WPNFs) combined with carvacrol (CA) and glycerol (Gly) were used to prepare edible coating to preserve Cheddar cheese by Wang et al., 2019 [91]. In general, WPNFs with CA and Gly films showed greater reducing power, better antimicrobial activity, and smoother surfaces with transparency values of 49.7%. Furthermore, cheddar cheese with WPNFs-CA/Gly coatings has shown lower weight losses and better textural properties than those in uncoated samples.

Potassium sorbate is the salt of sorbic acid, which restricts the activity of fungi, yeasts, and aerophilic bacteria. Potassium sorbate ionizes, forming sorbic acid when dissolved in water. At pH of 6.5, it is very effective, and as the pH decreases, its effectiveness increases. It is more effective when in non-dissociated form due to its greater ability to penetrate the cytoplasmic membrane of bacteria [82]. The incorporation of potassium sorbate into edible WPC films with two different pH values was tested by Perez et al. [82]. The authors studied the antimicrobial potential of films containing potassium sorbate by placing 12 mm diameter discs on plates containing 10 mL of Mueller-Hinton agar broth at pH 5.2, previously seeded with each bacterial suspension. Control films were also prepared with pH 5.2 and 6 but without containing bacterial suspension. It was verified that the addition of potassium sorbate in the WPC films inhibited the growth of non-O157 Shiga Toxin-producing *E. coli* isolated strains.

**Table 3 foods-13-02638-t003:** Examples of the application of various additives and their plasticizers in whey protein coatings and films.

Additives	Coatings/Films	Plasticizer	Main Properties	Reference
PREBIOTICS AND PROBIOTICS				
*Lactobacillus* *paracasei* CIDCA 8339 and *Kluyveromyces marxianus* CIDCA 8154	Whey proteins and kefiran	Glycerol	Probiotic activity	[92]
*Lactobacillus curvatus* 54M16	Whey protein/inulin/gelatin	Glycerol	Antimicrobial activity	[93]
*Bifidobacterium animalis Bb-12^®^* and *Lactobacillus casei-01*	WPI	Glycerol	Antimicrobial activity, probiotic activity	[42]
*Inulin*, *fructooligosaccharides*, *Bifidobacterium animalis* subsp. *lactis BB-12*	WPI and alginate	Glycerol	Probiotic activity	[94]
*Lactobacillus casei*	WPI	Glycerol	Anti-ripening process	[95]
*Lactic acid bacteria* (*LAB*), *Lactobacillus buchneri*	Protein-based films and coatings	Glycerol	Antifungal properties	[96]
ESSENTIAL OILS				
Almond and walnut oil	WPI	Glycerol	Emulsified films showed a more hydrophobic nature, decrease the water vapor permeability and increase the contact angle values.	[24]
Oregano, garlic oils nisin, and natamycin	WPI	Glycerol and Candelilla wax	Antimicrobial activity	[97]
*Laurus nobilis* L. and *Salvia officinalis*	WPI	Glycerol	antioxidant properties	[79]
Oregano (*Origanum vulgare* L.)	WPC and *Lepidium perfoliatum* L. gum (LPG)	Glycerol	Antimicrobial and antioxidant properties	[28]
Oregano (*Origanum compactum*) and Salvia (*Salvia sclarea*)	Sheep’s second cheese whey (SCW)	Glycerol	Protective coating cheese	[83]
Thyme (*Thymus vulgaris*)	Tamarind starch and WPC	Glycerol	Antimicrobial Agents.	[98]
Lemon and bergamot	WPI	Glycerol	Antimicrobial activity.	[81]
Tarragon (*Artemisia dracunculus*)	WPI	Glycerol	Antimicrobial properties	[99]
Oregano, clove, tea tree, coriander, mastic thyme, laurel, rosemary, and sage	WPI	Glycerol	Antimicrobial activity	[100]
*Cinnamomum cassia*, *Cinnamomum zeylanicum*, *Rosmarinus officinalis*	WPC	Glycerol	Antioxidant	[101]
PLANT EXTRACTS				
Curcumin	WPI	Glycerol	Antiviral, antioxidant, anti-infection, and antimicrobial activities	[102]
*Urtica dioica* L.	Whey protein isolate (WPI)	Glycerol	Antioxidant and antibacterial	[103]
Tarbush (*Fluorensia* *cernua*)	Whey protein	Glycerol and candelilla wax	Inhibiting the growth of pathogenic fungi	[104]
Fireweed extract (*Epilobium angustifolium* L.)	Gelatin and WPI	Glycerol	Phytoantioxidants	[105]
Yerba mate and white tea	Furcellaran/WPI	Glycerol	Antioxidant properties	[106]
Rosemary and sage extracts	WPC	Glycerol	Antioxidant activity	[107]
Garlic (*Allium sativum*) and white pepper (*Piper nigrum* L.)	WPI	Glycerol	Storage stability	[108]
Green tea extract	WPC	Glycerol	Retarding the lipid oxidation	[109]
OTHER ADDITIVES				
Liquid smoke	WPC	Glycerol	Antimicrobial, colouring, and flavouring properties	[110]
Spent coffee grounds	Whey protein	Glycerol	Antioxidants	[111]
γ-Aminobutyric acid (GABA)	WPI	Glycerol	Regulates blood pressure and insulin, protects the nervous system, fights diabetes and cancer	[37]
Liquid smoke	WPC	Glycerol	Antimicrobial and preservation	[112]

## 6. Future Perspectives

The use of films and coatings to improve the organoleptic properties of foods and extend the shelf life of these products is one of the industry’s concerns. Consumers are increasingly demanding in relation to the products they consume, both nutritionally and aesthetically. Therefore, one of the industry’s challenges is to obtain edible films and coatings that meet all consumer demands and bring economic and environmental benefits.

Protein-based films and coatings offer good sustainability and versatility in different sectors. They can be applied as food packaging, in pharmaceutical and medical areas, in agriculture, in personal hygiene products, and as intelligent packaging with a strong positive impact on the environment. Several studies must continue to be carried out to improve the properties and functionality and its potential for application in edible coatings and films [67]. However, other improvements must be performed [113], namely regarding the improvement of the hydrophobicity of films obtained with natural biopolymers and whey proteins in order to increase water vapor transmission [114].

Previous works indicate the potential of whey-based coatings to serve other functions (i.e., carrier of antimicrobials, antioxidants, or other nutraceuticals) without significantly compromising the desirable primary barrier and mechanical properties of packaging films [10,44].

More recently, two reviews focusing on the use of milk or whey protein-based films and coatings present several examples of the application of such films to different foods. Kandasamy et al., 2021 [10] refer that WP-based films/coatings outperform synthetic plastics in terms of intrinsic edibility and biodegradability. Exceptional optical and barrier properties that surpass existing biopolymers can also be attained with whey-based biopolymers. WP coatings have been proven as efficient gas barriers capable of acting as vehicles for several compounds that include antioxidants, antimicrobials, or different nutrients, although their mechanical properties need to be improved. Similar conclusions are also presented by Chaudhary et al., 2022 [115]. The addition of new antimicrobial, antioxidant, and anti-darkening agents and nanoparticles, the combination of film-forming biopolymers, and the incorporation of plasticizers are alternatives to benefit whey-based edible films and coatings. The application of commercial bioprotective cultures in whey-based films and coatings matrix is clearly one opportunity to potentiate the use of whey as a base material for packaging.

However, despite all the developments made so far to obtain efficient, functional, and sustainable packaging, there are still very important issues that must also be further investigated. One of the problems faced so far is the amplification of these packaging materials made with natural biopolymers at an industrial level and also in real time [116]. In fact, at the industrial level, the production of whey protein-based films may pose some challenges. Firstly, the heat denaturation process may cause the coagulation of whey proteins and the formation of a coarse gel. Thus, the process must be performed by heating under high-shear conditions. The continuous recirculation of the formulation on a high-pressure homogenizer while heating may allow for the obtention of a viscous liquid capable of forming the coating. Secondly, the prepared films can allow for the growth of microorganisms and, therefore, have a short shelf-life under refrigeration conditions, which can be a problem for logistics and distribution. The use of preservatives can be a solution but may not be well accepted by consumers. The product can be UHT sterilized and aseptically packaged, but this solution implies expensive equipment. It is also possible to process whey-based formulations for packaging applications and edible coatings through extrusion as well as compression molding. Research has shown the importance of suitable conditions to control denaturation and cross-linking as well as the benefits of whey mixed with plant proteins [117].

Another challenge associated with the industrial application of whey-based films and coatings is related to consumers’ acceptance of them. Although in our previous works based on the application of whey-based coatings to cheeses, the sensory characteristics of the product were not affected, the study of Zhang et al. [118] indicates that although edible food packaging helps address the problem of waste generated by conventional packaging, consumers’ acceptance of this novel practice remains uncertain. Consumers are often skeptical when adopting new sustainable products, which results in attitudinal ambivalence. The authors suggested strategies for producers and policymakers to improve their communication with consumers regarding the introduction of new edible packaging materials into the market.

## 7. Conclusions

Edible films and coatings offer a sustainable and versatile solution to various environmental problems and benefit human health. Furthermore, edible films and coatings prepared from renewable sources have been widely explored due to their cost-effectiveness, compatibility, non-toxicity, and biodegradability. The materials used to obtain films and coatings, such as polysaccharides, proteins, and lipids, determine their processability and film formation, casting, lamination, and other important properties. Blends of biopolymers, composites, complex multilayers, and the use of antimicrobial agents, antifungals, enzymes, vitamins, essential oils, and plasticizers allow new techniques for developing edible films and coatings.

Furthermore, active and intelligent packaging and the use of new nanotechnologies are trends for application in edible food packaging. The use of agri-food by-products, such as cheese whey, as sources of materials for edible films and coatings represents a promising alternative for the valorization of this product that represents roughly eight times the amount of cheese produced worldwide and has a harmful environmental impact if not treated properly.

Whey protein films prove to be excellent vehicles for the delivery of functional and bioactive compounds in foods, helping to extend shelf life and safety and improving the nutritional properties as well as the sensory qualities of these foods. Therefore, continuous research and development efforts aimed at improving the functional and mechanical properties of such materials are a hot topic. Furthermore, the use of edible whey protein films and coatings developed from biological materials in a circular economy perspective can reduce food waste and minimize the environmental impacts that plastic packaging causes to the environment. Therefore, this potential deserves greater research efforts.

## Figures and Tables

**Figure 1 foods-13-02638-f001:**
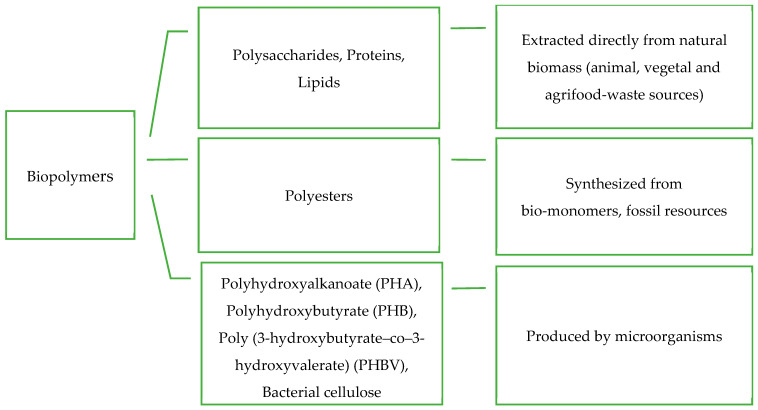
Scheme of the main biopolymers classification and their origin [17].

**Figure 2 foods-13-02638-f002:**
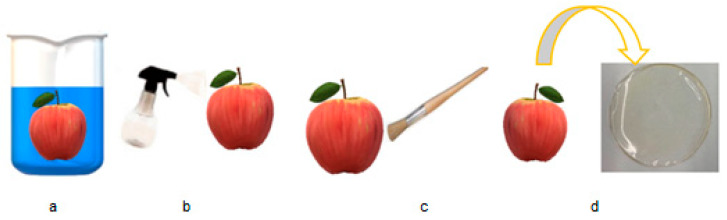
Examples of the application of coatings and films: (**a**) dipping, (**b**) spraying, (**c**) brushing, and (**d**) wrapping.

**Figure 3 foods-13-02638-f003:**
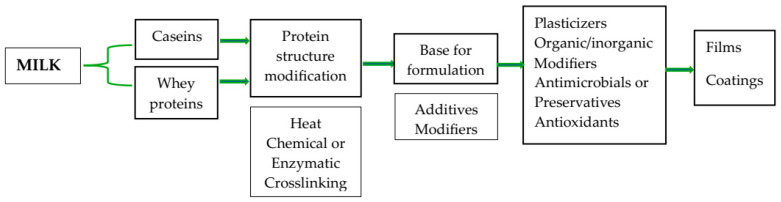
Example of production of films and coatings with milk proteins.

**Table 1 foods-13-02638-t001:** Main biopolymers used and functionality in the manufacture of edible films and coatings [21,22,27].

Materials	Example of Materials	Properties	Functionality
Polysaccharides	Alginate, Pectin, Cellulose, Starch, Chitosan, Agar	Thickeners Gellants Emulsifiers Stabilizers Coating	Polysaccharides form the structure of a solid polymer matrix and improve gas barrier properties.
Proteins	Whey Protein, Gelatin, Casein, Collagen, Ovalbumin Soy Protein	Gellants Thickeners Stabilizers Foaming Firmness Coating	Transport of antimicrobials and antioxidants. They control the transport of gases (oxygen).
Lipids	Waxes Paraffin Glycerides	Coating	Avoid drying or dehydration and give flexibility. Prevent moisture migration and water vapor transmission.

**Table 2 foods-13-02638-t002:** Mechanical properties (TS, EAB, YM, WVP, and thickness) of different film protein sources [17,46,49,50].

Protein Source	TS (MPa)	EAB (%)	YM (MPa)	WVP (g·mm/m^2^ h kPa)	Thickness (μm)
Soy *	1.59–8.05	105.3	87.4	-	82
Zein	4	118	-	4	-
Milk Casein	2–77	2–130	-	1.6–11	-
WPC	0.7–0.9	18	-	6.2–12.8	132
WPI	0.9–8.20	33–72.4	24.71	4.57	119.86
Grass Pea *	8.59	68.3	483.0	-	110

TS (MPa): tensile strength; EAB: elongation at break; YM: Young’s modulus; WVP: water vapor permeability; WPC: whey protein concentrate; WPI: whey protein isolate. * (at pH 7).

## Data Availability

No new data were created or analyzed in this study.

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
