# Peer review of "A Review of Recent Developments in Edible Films and Coatings-Focus on Whey-Based Materials"

_foods, 2024, doi:10.3390/foods13162638_

Round 1

Reviewer 1 Report

Comments and Suggestions for Authors

This paper is a very valuable material for publication with a lot of literature and data searched, but it needs to be reorganized in order to highlight its full importance and capacity to become a reference for future manuscripts. The topic of this manuscript are edible biopolymer films and coatings with a focus on whey based materials. In order to highlight the part related to whey based materials in the work, the first part of the manuscript (section 2. - 2.4.) needs to be reworked and shortened. There are many review papers in the literature dealing with edible films and coatings, and what can highlight your work and give it additional value is focus on whey based materials. Currently subheadings are listed as:

1. Introduction

2. Edible Fims and Coatings

2.1. Films and coatings in foods.

2.2. Films and coatings’ properties.

2.3. Biopolymer-based films and coatings.

2.3.1. Polysaccharide based films and coatings.

2.3.2. Lipid based films.

2.3.3. Composite and nanocomposite films and coatings.

2.3.4. Protein based films.

2.4. Methods to obtain films and coatings.

2.5. Whey protein films.

2.5.1. Methods to obtain whey protein films and coatings.

3. Advantages and disadvantages of protein films and coatings.

4. Improvement of protein film’s properties.

5. Plasticizers.

5.1, Effect of plasticizer type on film properties.

5.2. Advantages and disadvantages of plasticizers in films and coatings.

6. Additives for functional films and coatings.

6.1. Prebiotics and/or probiotics incorporated in edible films and coatings.

6.2. Antioxidants, antibacterial and antifungal compounds.

Future perspectives.

7. Conclusion.

 I suggest the following subheadings model:

1. Introduction

2. Biopolymer-based films and coatings (where you shortly present hydrocolloids, lipids and composites)

3. Edible Fims and Coatings properties for food packaging application (where you point to the main properties, advantages and disadvantages)

4. Whey protein films

4.1. Methods to obtain whey protein films and coatings (only in this paragraph explain how films and coating are obtained. In previous subheading model you had two subheading dealing with the “Methods to obtain films and coatings” and “Methods to obtain whey protein films and coatings”. Merge texts into one subheading where the introduction is about obtaining films and coatings in general, and the main text is about obtaining whey films and coatings)

4.2. Advantages and disadvantages of whey films and coatings

4.3. Improvement of wey film’s properties: additives for functional films and coatings

4.3.1. Effect of plasticizers on film properties

4.3.2. Effect of prebiotics and/or probiotics incorporated in edible films and coatings

4.3.3. Effect of antioxidants, antibacterial and antifungal compounds

5. Future perspectives

6. Conclusion

 Additional specific comments:

1. line 78-79: change “…consumed together with food products, without the need to remove them.” to “…consumed together with food products, without the need to be removed.”

Line 81-82: “while minimize environmental problems that other packaging creates. [1,16,17].” Change to “…with minimal envoronmental impact.”

2. table 1: functionality of polysaccharides is “Polysaccharides form the structure of a solid polymer matrix”? This part is not clear.

3. line 490: “and less permeable (Figure 4).” Figure 4 is missing in the manuscript.

4. line 605-608: “In some cases, water molecules can also act as plasticizers in starch films, causing a phenomenon known as retrogradation, which is a rearrangement of amorphous starch chains in the presence of moisture during storage.” Delete this part since it has no relation to whey materials.

Line 703-713: delete this paragraph since this is not an example of using whey based materials with eutectic solvents.

Line 714-725: could you find application of these solvents with whey based films? If you find it replace this paragraph. If you don’t find it keep this text since it is related to protein isloates.

Line 850-856: delete this part since it has no relation to whey materials.

Line 868-874: delete this part since it is not related to the toppic of your manuscript. Try to find example where carvacrol was used in combination with whey based films.

Line 884-893: delete this part because it is not relevant to the topic of your manuscript.

Future perspectives: direct the text to whey based films and coating. As it is now, it is too general and applies to the entire field of biopolymer materials.

 Comments on the Quality of English Language

This paper is a very valuable material for publication with a lot of literature and data searched, but it needs to be reorganized in order to highlight its full importance and capacity to become a reference for future manuscripts. The topic of this manuscript are edible biopolymer films and coatings with a focus on whey based materials. In order to highlight the part related to whey based materials in the work, the first part of the manuscript (section 2. - 2.4.) needs to be reworked and shortened. There are many review papers in the literature dealing with edible films and coatings, and what can highlight your work and give it additional value is focus on whey based materials. Currently subheadings are listed as:

1. Introduction

2. Edible Fims and Coatings

2.1. Films and coatings in foods.

2.2. Films and coatings’ properties.

2.3. Biopolymer-based films and coatings.

2.3.1. Polysaccharide based films and coatings.

2.3.2. Lipid based films.

2.3.3. Composite and nanocomposite films and coatings.

2.3.4. Protein based films.

2.4. Methods to obtain films and coatings.

2.5. Whey protein films.

2.5.1. Methods to obtain whey protein films and coatings.

3. Advantages and disadvantages of protein films and coatings.

4. Improvement of protein film’s properties.

5. Plasticizers.

5.1, Effect of plasticizer type on film properties.

5.2. Advantages and disadvantages of plasticizers in films and coatings.

6. Additives for functional films and coatings.

6.1. Prebiotics and/or probiotics incorporated in edible films and coatings.

6.2. Antioxidants, antibacterial and antifungal compounds.

Future perspectives.

7. Conclusion.

 I suggest the following subheadings model:

1. Introduction

2. Biopolymer-based films and coatings (where you shortly present hydrocolloids, lipids and composites)

3. Edible Fims and Coatings properties for food packaging application (where you point to the main properties, advantages and disadvantages)

4. Whey protein films

4.1. Methods to obtain whey protein films and coatings (only in this paragraph explain how films and coating are obtained. In previous subheading model you had two subheading dealing with the “Methods to obtain films and coatings” and “Methods to obtain whey protein films and coatings”. Merge texts into one subheading where the introduction is about obtaining films and coatings in general, and the main text is about obtaining whey films and coatings)

4.2. Advantages and disadvantages of whey films and coatings

4.3. Improvement of wey film’s properties: additives for functional films and coatings

4.3.1. Effect of plasticizers on film properties

4.3.2. Effect of prebiotics and/or probiotics incorporated in edible films and coatings

4.3.3. Effect of antioxidants, antibacterial and antifungal compounds

5. Future perspectives

6. Conclusion

Additional specific comments:

1. line 78-79: change “…consumed together with food products, without the need to remove them.” to “…consumed together with food products, without the need to be removed.”

Line 81-82: “while minimize environmental problems that other packaging creates. [1,16,17].” Change to “…with minimal envoronmental impact.”

2. table 1: functionality of polysaccharides is “Polysaccharides form the structure of a solid polymer matrix”? This part is not clear.

3. line 490: “and less permeable (Figure 4).” Figure 4 is missing in the manuscript.

4. line 605-608: “In some cases, water molecules can also act as plasticizers in starch films, causing a phenomenon known as retrogradation, which is a rearrangement of amorphous starch chains in the presence of moisture during storage.” Delete this part since it has no relation to whey materials.

Line 703-713: delete this paragraph since this is not an example of using whey based materials with eutectic solvents.

Line 714-725: could you find application of these solvents with whey based films? If you find it replace this paragraph. If you don’t find it keep this text since it is related to protein isloates.

Line 850-856: delete this part since it has no relation to whey materials.

Line 868-874: delete this part since it is not related to the toppic of your manuscript. Try to find example where carvacrol was used in combination with whey based films.

Line 884-893: delete this part because it is not relevant to the topic of your manuscript.

Future perspectives: direct the text to whey based films and coating. As it is now, it is too general and applies to the entire field of biopolymer materials.

 Author Response

REVIEWER 1

This paper is a very valuable material for publication with a lot of literature and data searched, but it needs to be reorganized in order to highlight its full importance and capacity to become a reference for future manuscripts. The topic of this manuscript are edible biopolymer films and coatings with a focus on whey based materials. In order to highlight the part related to whey based materials in the work, the first part of the manuscript (section 2. - 2.4.) needs to be reworked and shortened. There are many review papers in the literature dealing with edible films and coatings, and what can highlight your work and give it additional value is focus on whey based materials.

Currently subheadings are listed as:

  1. Introduction
  2. EdibleFimsand Coatings

2.1. Films and coatings in foods.

2.2. Films and coatings’ properties.

2.3. Biopolymer-based films and coatings.

2.3.1. Polysaccharide based films and coatings.

2.3.2. Lipid based films.

2.3.3. Composite and nanocomposite films and coatings.

2.3.4. Protein based films.

2.4. Methods to obtain films and coatings.

2.5. Whey protein films.

2.5.1. Methods to obtain whey protein films and coatings.

  1. Advantagesanddisadvantages of protein films and coatings.
  2. Improvementofprotein film’s properties.
  3. Plasticizers.

5.1, Effect of plasticizer type on film properties.

5.2. Advantages and disadvantages of plasticizers in films and coatings.

  1. Additivesforfunctional films and coatings.

6.1. Prebiotics and/or probiotics incorporated in edible films and coatings.

6.2. Antioxidants, antibacterial and antifungal compounds.

Future perspectives.

  1. Conclusion.

I suggest the following subheadings model:

  1. Introduction
  2. Biopolymer-basedfilmsand coatings (where you shortly present hydrocolloids, lipids and composites)
  3. EdibleFimsand Coatings properties for food packaging application (where you point to the main properties, advantages and disadvantages)
  4. Wheyproteinfilms

4.1. Methods to obtain whey protein films and coatings (only in this paragraph explain how films and coating are obtained. In previous subheading model you had two subheadings dealing with the “Methods to obtain films and coatings” and “Methods to obtain whey protein films and coatings”. Merge texts into one subheading where the introduction is about obtaining films and coatings in general, and the main text is about obtaining whey films and coatings)

4.2. Advantages and disadvantages of whey films and coatings

4.3. Improvement of wey film’s properties: additives for functional films and coatings

4.3.1. Effect of plasticizers on film properties

4.3.2. Effect of prebiotics and/or probiotics incorporated in edible films and coatings

4.3.3. Effect of antioxidants, antibacterial and antifungal compounds

  1. Futureperspectives
  2. Conclusion

ANSWER:

Thank you for your comments and suggestions to alter sequence of the presentation and subheadings. The subheadings were altered as suggested. This is a major improvement to the readability of the manuscript.

Additional specific comments:

  1. line 78-79: change “…consumed together with food products, without the need to remove them.” to “…consumed together with food products, without the need to be removed.”

Altered as suggested. (See lines 81-85)

Line 81-82: “while minimize environmental problems that other packaging creates. [1,16,17].” Change to “…with minimal envoronmental impact.”

Altered as suggested. (See lines 81-85)

  1. table 1: functionality of polysaccharides is “Polysaccharidesform the structure of a solidpolymer matrix”? This part is not clear.

Altered as suggested. Text was made clearer.

  1. line 490: “andlesspermeable (Figure 4).” Figure 4 is missing in the manuscript.

Altered as suggested. The reference to figure 4 was eliminated.

  1. line 605-608: “In some cases, water molecules can also act as plasticizers in starch films, causing a phenomenon known as retrogradation, which is a rearrangement of amorphous starch chains in the presence of moisture during storage.” Delete this part since it has no relation to whey materials.

Eliminated as suggested.

Line 703-713: delete this paragraph since this is not an example of using whey based materials with eutectic solvents.

Eliminated as suggested.

Line 714-725: could you find application of these solvents with whey based films? If you find it replace this paragraph. If you don’t find it keep this text since it is related to protein isloates.

Eliminated as suggested.

Line 850-856: delete this part since it has no relation to whey materials.

Eliminated as suggested.

Line 868-874: delete this part since it is not related to the toppic of your manuscript. Try to find example where carvacrol was used in combination with whey-based films.

Eliminated as suggested.

Line 884-893: delete this part because it is not relevant to the topic of your manuscript.

Eliminated as suggested.

Future perspectives: direct the text to whey-based films and coating. As it is now, it is too general and applies to the entire field of biopolymer materials.

Revised. (See lines 742-769).

Reviewer 2 Report

Comments and Suggestions for Authors

Thank you very much for the opportunity to read the authors' point of view on the subject of edible biofilms. It is extremely up-to-date and fits into the scientific interests of technologists. Such a synthetic study on this topic is needed, but the authors still need to work on it. Valuable information, but presented in a very chaotic way. Many of them are repeated in different places in the article, so the text needs to be tidied up and edited a bit. In review articles, it is good to write general things first and then go into detail. Below I have included my comments on the text (some may result from imperfections in the translation, but should be clarified):

- NOTE TO THE ENTIRE TEXT: please do not put dots at the end of the titles of subsections and chapters of the text
- lines 52-53: What does this sentence mean: "Plastics also impose a series of risks and health problems when they enter in the food chain..." Do the authors think that plastics are currently absent from the food chain? It's not true. Please explain what the authors meant here?
- lines 84-85 - this is one of the examples of repetitions of the information that appeared earlier about how to obtain films. Similarly, lines 146-148 repeat the information previously provided. You should read the entire text and move fragments to the appropriate places in the text so as not to repeat yourself or repeat information in the text that is also included in the tables (e.g. No. 1).
- lines 97-98: where did the abbreviations "WPI" and "WPC" suddenly come from - I hadn't noticed them in the text before.

- lines 104-108: a very short fragment about the disadvantages of films, so when in line 466 (chapter 3) I saw that the authors were writing about the advantages and disadvantages again, but only about protein films and coatings, I was surprised. So please put all the advantages and disadvantages of all varieties of films and coatings in one fragment.

- line 117: title of chapter 2.1. it does not contain the content described therein at all. That's why I propose changing it.
- line 127 - the title needs to be worded differently, because it's about crafting, so it doesn't match the current one.
- line 148: suddenly the concept of "encapsulation" appeared, which was not preceded by any introduction - you need to justify the introduction of this term.
- line 154: the phrase "masking flavours" has a very negative connotation. Indicates adulteration of an undesirable odor. Did the authors use it in such a negative context?

- line 208 - abbreviation "WVP" - I can't tell what it means from the description. Please explain in more detail.
- lines 223-224: repetition again, additionally it was in table 1.
- lines 222 and 231: repetition of information.
- line 272: "The source and type of lipids used..." but in what, for what?
- line 285: Title 2.3.3 - where did the idea for such a subtitle come from? It does not fit the concept presented earlier! It doesn't fit here at all.
- line 307: which material is toxic in this sentence?
- lines 312-314: repetition of the messages included in table 1.

- chapter 2.4 - I think it should be more extensive. Maybe it should be combined with subsection 2.4?
- line 580: The authors did not indicate the topic of "Plasticizers" before (only in figure 3) and suddenly a whole chapter about this issue. Is it necessary enough to devote so much space in a review article?

- line 671:"Peres et al." - without the list item number, which appeared only at the end of paragraph "[66]" (line 680).
- line 681: "...by Nuanmano and coworkers..." - reference to the number only in line 688. This should be included right after the authors' name.
- lines 811-817: reference number 35 is mentioned three times in one paragraph. Only once, at the end of the paragraph, would be enough.
- Table No. 3: I think it is unnecessary because all the information it contains is included in the text.
- line 896: title "Future perspectives", but what are they about?
- List of references: the list contains 121 references, while the last one I found in the text was item number 119. Therefore, numbers 120 and 121 are missing in the text.

Author Response

REVIEWER 2 

 Thank you very much for the opportunity to read the authors' point of view on the subject of edible biofilms. It is extremely up-to-date and fits into the scientific interests of technologists. Such a synthetic study on this topic is needed, but the authors still need to work on it. Valuable information, but presented in a very chaotic way. Many of them are repeated in different places in the article, so the text needs to be tidied up and edited a bit. In review articles, it is good to write general things first and then go into detail. Below I have included my comments on the text (some may result from imperfections in the translation, but should be clarified):

- NOTE TO THE ENTIRE TEXT: please do not put dots at the end of the titles of subsections and chapters of the text.

Corrected.
- lines 52-53: What does this sentence mean: "Plastics also impose a series of risks and health problems when they enter in the food chain..." Do the authors think that plastics are currently absent from the food chain? It's not true. Please explain what the authors meant here? Sentence was made clearer. Sentence from references 8 and 9. (see lines 53-55)
- lines 84-85 - this is one of the examples of repetitions of the information that appeared earlier about how to obtain films. Similarly, lines 146-148 repeat the information previously provided. You should read the entire text and move fragments to the appropriate places in the text so as not to repeat yourself or repeat information in the text that is also included in the tables (e.g. No. 1).

Repetitions were eliminated.
- lines 97-98: where did the abbreviations "WPI" and "WPC" suddenly come from - I hadn't noticed them in the text before.

Corrected. First time abbreviations appear are explained.

- lines 104-108: a very short fragment about the disadvantages of films, so when in line 466 (chapter 3) I saw that the authors were writing about the advantages and disadvantages again, but only about protein films and coatings, I was surprised. So please put all the advantages and disadvantages of all varieties of films and coatings in one fragment.

Altered as suggested and sentence eliminated.

- line 117: title of chapter 2.1. it does not contain the content described therein at all. That's why I propose changing it.

All the sequence of information presented according to the proposal for subheadings presented by reviewer 1.
- line 127 - the title needs to be worded differently, because it's about crafting, so it doesn't match the current one.

Altered
- line 148: suddenly the concept of "encapsulation" appeared, which was not preceded by any introduction - you need to justify the introduction of this term.

Information deleted.
- line 154: the phrase "masking flavours" has a very negative connotation. Indicates adulteration of an undesirable odor. Did the authors use it in such a negative context?

Information deleted.

- abbreviation "WVP" - I can't tell what it means from the description. Please explain in more detail.

Explanation introduced. (See lines 308,309)
- lines 223-224: repetition again, additionally it was in table 1.

Information deleted.
- lines 222 and 231: repetition of information.

Information deleted
- line 272: "The source and type of lipids used..." but in what, for what?

Clarified. (See line 195)
- line 285: Title 2.3.3 - where did the idea for such a subtitle come from? It does not fit the concept presented earlier! It doesn't fit here at all.

We consider that composite and nanocomposite films and coatings should also be referred to, since these types of materials are “upgrades” of conventional formulations based on just one material.
- line 307: which material is toxic in this sentence?

Explained on lines 249-251.
- lines 312-314: repetition of the messages included in table 1.

Information deleted.

- chapter 2.4 - I think it should be more extensive. Maybe it should be combined with subsection 2.4?

All the sections were altered, and methods were included in other section.
- line 580: The authors did not indicate the topic of "Plasticizers" before (only in figure 3) and suddenly a whole chapter about this issue. Is it necessary enough to devote so much space in a review article?

Information about plasticizers was reduced and included in the section referring to film forming materials. Is now inserted on section 2.2. (see lines 261-302)

- line 671:"Peres et al." - without the list item number, which appeared only at the end of paragraph "[66]" (line 680).

Corrected. (See lines 736-738)
- line 681: "...by Nuanmano and coworkers..." - reference to the number only in line 688. This should be included right after the authors' name.

Eliminated
- lines 811-817: reference number 35 is mentioned three times in one paragraph. Only once, at the end of the paragraph, would be enough.

Corrected. (See lines 603-613)
- Table No. 3: I think it is unnecessary because all the information it contains is included in the text.

Table 3 intends to summarize information discussed in the text. Besides, most of the information included in table 3 was not previously indicated in the text (references 89 to 109).

- line 896: title "Future perspectives", but what are they about?

Revised. (See lines 742-769).

- List of references: the list contains 121 references, while the last one I found in the text was item number 119. Therefore, numbers 120 and 121 are missing in the text.

List of references revised and corrected. It has now 112 references.

Reviewer 3 Report

Comments and Suggestions for Authors

Very high percentage of plagiarism is detected with words and phrases copy pasted from previously published literature and internet sources. Authors need to do major revisions to minimize the plagiarism.

Author Response

Document was tested in a specific program and level of similitude to the sources is low. Please see attached report. Most of the information indicated as plagiarism, in fact is related to the references list.

Round 2

Reviewer 1 Report

Comments and Suggestions for Authors

I want to thank the authors for considering all the comments and making changes to the manuscript. I propose the manuscript in this form for publication.

Author Response

Thank you very much. With your help we think that the quality of the manuscript was clearly improved.

Reviewer 3 Report

Comments and Suggestions for Authors

This review article discusses advancements in food packaging, emphasizing whey protein-based films and coatings. These biopolymers offer environmental benefits, biodegradability, and nutritional advantages over traditional plastics. The article examines the physicochemical properties of whey protein films, their mechanical resistance, gas barrier capabilities, and combinations with other polymers. The focus is on improving these properties through new methodologies and molecular-level manipulation to enhance the functionality and broaden the applications of these biopolymer films in food packaging. Kindly address minor comments prior to publication.

1. How do whey protein-based films compare with other biopolymer films in terms of cost and performance?

2. What are the challenges associated with scaling up the production of whey protein-based films?

3. Can you provide specific examples or data on the shelf life extension of food products using whey protein-based films?

4. How do the mechanical properties of whey protein films change under different humidity levels?

5. How significant are the nutritional benefits of using whey protein films in real-world applications?

6. Have you explored the antimicrobial properties of whey protein films?

7. Are there any studies or data on consumer acceptance of edible films and coatings, particularly those based on whey proteins?

Author Response

Thank you very much for your help. We tried to address all the questions raised.
